# Endogenous Hormones and Biochemical Changes during Flower Development and Florescence in the Buds and Leaves of *Lycium ruthenicum* Murr

**Youyan Guo** [1,2], **Lizhe An** [1,*], **Hongyuan Yu** [2] **and Miaomiao Yang** [3]

1   Ecology College, Lanzhou University, South Tianshui Road No.222, Lanzhou 730000, China; guoyouyan_2008@163.com

2   Architecture and Construction College, Hexi University, North Loop Road No.846, Zhangye 734000, China; yhy_2000_113@163.com

3   Life Sciences and Food Engineering College, Shaanxi Xueqian Normal University, Chang'an District Shenhe Two Road No.101, Xi'an 710000, China; yangmmgg@163.com

\*   Correspondence: lizhean2018@163.com; Tel.: +86-189-1988-9999

**Abstract:** *Lycium ruthenicum* Murr. is one of the most important shrubs grown in northwest China. Healthy buds and leaves of *L. ruthenicum* Murr. were selected for the present study. Flower development was divided into six stages, namely, flower bud pre-differentiation (I), late flower differentiation (II), squaring stage (III), dew crown period (IV), open stage (V) and senescent stage (VI). Endogenous hormone content and specific value, soluble sugar, sucrose, starch, and soluble protein were measured, and ABA/IAA, ABA/GA$_3$, ZR/IAA, ZR/GA$_3$, and C/N were calculated in buds and leaves at stage VI. The results showed that ABA, GA$_3$, and ZR content of buds significantly increased from flower bud pre-differentiation to late flower differentiation stage. However, ABA, GA$_3$, and ZR content of leaves had the opposite change trend. From open stage to senescent stage, IAA, ABA, and GA$_3$ content of buds and leaves significantly increased in *L. ruthenicum* Murr. However, ZR content of buds and leaves significantly decreased from open stage to senescent stage. ABA/IAA, ABA/GA$_3$, ZR/IAA, and ZR/GA$_3$ values of buds significantly increased from lower bud pre-differentiation to late flower differentiation stage. However, ABA/IAA, ABA/GA$_3$, ZR/IAA, and ZR/GA$_3$ values of leaves significantly decreased from lower bud pre-differentiation to late flower differentiation stage. ABA/IAA and ABA/GA$_3$ of buds significantly increased from open stage to senescent stage, but ZR/IAA and ZR/GA$_3$ of buds significantly decreased from open to senescent. At this stage, ABA/IAA, ABA/GA$_3$, ZR/IAA, and ZR/GA$_3$ significantly decreased in *L. ruthenicum* Murr. The higher soluble sugar and sucrose content in the buds and leaves were beneficial to the flower bud differentiation of *L. ruthenicum*. The increasing of soluble sugar improved the energy basis to florescence and senescent. The carbohydrates metabolism enhanced from open stage to senescent stage and nitrogen metabolism reduced from open stage to senescent stage of *L. ruthenicum*.

**Keywords:** flower differentiation; flower development; florescence; endogenous hormones; carbohydrates; soluble protein

## 1. Introduction

    *Lycium ruthenicum* Murr. is a perennial deciduous shrub of the Solanaceae family. The plant is distributed mainly across northwest China. Its fruit is rich in anthocyanins, polysaccharides, and proteins, has positive effects on eyesight as an anti-hypertensive, and lowers cholesterol [1,2]. With the increase in interest to *L. ruthenicum*, natural vegetation was seriously damaged. The population of *L. ruthenicum* has dropped in recent years. Thus, research on a key link plant breeding system is important to *L. ruthenicum* survival.

    Flower development is the key link in plant breeding system and a highly complex process that is characterized by two distinct physiological phases: (a) bud initiation and

(b) floral bud development [3]. Flowering has a correlation with the endogenous hormone levels. The ABA was controversial during the floral transitions, as both positive and negative effects of ABA have been reported [4]. IAA was also controversial [5]. Carbohydrate and nitrogen content had the important role of flower development, which were the basis of flower development. Florescence is an important component of plant breeding systems [6]. The flowering phase of *L. ruthenicum* takes 5–9 months to complete. Its flowers are bisexual and have purple and pink petals and clearly visible veins. Sexual reproduction is an important period in *L. ruthenicum* growth and development and has a crucial influence on *L. ruthenicum* fruit yield. Florescence is controlled by complex physiological processes [7] and involves internal and external factors. External factors include light, temperature, water, and nutrients, while internal factors include hormones, carbohydrates, and nutrient substances [8,9].

Researchers have investigated the relationship between flower development and hormones, and the physiology and biochemistry in different fruit types. However, similar research papers did not include *L. ruthenicum.*

Therefore, the aims of this study were to clarify the changes in endogenous hormones and metabolites in the leaves and buds of *L. ruthenicum* and assess the effects of these concentrations on flower development. We will reveal the regulatory mechanism of flower development in *L. ruthenicum* and provide the theoretical basis on *L. ruthenicum* flowering phase management.

## 2. Materials and Methods

### 2.1. Plant Material

The floral buds of *L. ruthenicum* grown in the agronomy practice base of Hexi University under natural conditions, an annual mean temperature of 6 °C, annual mean evaporation of 2291 mm, and mean rainfall of 113–120 mm, were selected for the present study.

Flower development and florescence were observed, and divided into six stages. These stages were designated as flower bud pre-differentiation (I), late flower differentiation (II), squaring stage (III), dew crown period (IV), open stage (V), and senescent stage (VI), the buds and leaves of *L. ruthenicum* were taken in April to June 2021 (Figure 1). Leaves were taken from buds at every stage. Hormones, soluble sugar, sucrose, starch, and soluble protein levels were measured in leaf and bud samples during flowering development.

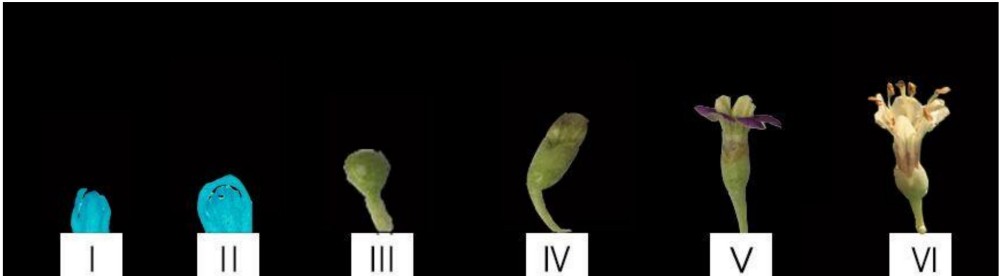

**Figure 1.** Flower development stages in *Lycium ruthenicum* Murr.

### 2.2. Hormone Analysis

Buds and leaves from the different developmental stages were immediately frozen in liquid nitrogen and stored at −80 °C for subsequent analyses. Endogenous hormones, including indole acetic acid (IAA), indolebutyric acid (IBA) abscisic acid (ABA), gibberellic acid ($GA_3$), and zeatin riboside (ZR), were measured by using high efficiency liquid chromatography (HPLC). Leaf and bud samples (0.4 g) were put in 10 mL centrifuge tube, and added to cold 80% methanol containing 0.5% methanoic acid at 4 °C in 2 mL. Ultrasonic extraction was performed by 30 min and let stand for the night. The extracts were centrifuged at 12,000× $g$ for 15 min. The residue added to a 2 mL precool extracting solution, and an ultrasonic extraction for 30 min was done. The methanol phase was reduced to an aqueous phase under reduced pressure on a rotary evaporator at 38 °C. After that, was incubated at

−20 °C refrigerator freeze for 30 min, and then were centrifuged at $12,000 \times g$ for 10 min. the supernatants were reserved and pigment and lipid were discarded. The purification pipe underwent severe concussion for 30 s, and then was centrifuged at $4000 \times g$ for 5 min. The supernatants were concentrated to near dry by pressure blowing concentrator, and then added to 1.0 mL acetonitrile solution. Ultrasonic extraction was performed by 30 min, and filtered by 0.22 μm Millipore filter. The filter liquor was detected by HPLC as described previously [10]. Every sample repeat measured three times, and calculated the average value.

## 2.3. Determination of Soluble Total Sugar and Sucrose

The 0.5 g buds or leaves were powdered, and then 5 mL of 80% ethanol was added. The mixture was incubated in boiling water for 30 min. Homogenates were centrifuged at $6000 \times g$ for 5 min. Residues were re-suspended in 3 mL of 80% ethanol and centrifuged at $6000 \times g$ for 5 min. The supernatants of the two obtained centrifugates were mixed and then added with 0.1 g of activated charcoal. The mixture was filtered, and the filtrate was used in the determination of soluble total sugar and sucrose. Total sugar and sucrose concentrations were evaluated at 630 nm via the spectrophotometer according to the anthrone method [11]. Soluble sugar was measured at 640 nm using methanol as blank. The concentration of soluble sugar was calculated using glucose solution as standard [12].

## 2.4. Starch Content

Approximately 0.5 g of the buds or leaves were powdered and then mixed with 6 mL of 80% ethanol. The mixture was incubated in boiling water for 30 min. The homogenates were centrifuged at $3000 \times g$ for 5 min. The supernatants were discarded, and 10 mL of 3 mol/L HCl was added to the sediment in boiling water for 45 min. Then, 10 mL 3 mol/L NaOH was added to the sediment. Around 2 mL of the sample solution was diluted to 10 mL, and warm 6 mL of 0.4% anthrone was added in a boiling water bath. Starch content was measured at 640 nm via the spectrophotometer according to the anthrone method [13].

## 2.5. Soluble Proteins

Soluble proteins were estimated by the method of Bradford [14]. A mixture of 0.3 g of buds and leaves was homogenized in 20 mL of 20% trichloroacetic acid (TCA), and then centrifuged at 800 rpm for 15 min. Supernatant was discarded. Pellets were mixed with 5 mL of 0.1 N NaOH to solubilize proteins, and solution was centrifuged again at 800 rpm for 15 min. The supernatant was mixed with 10 mL of 0.1 N NaOH and used for estimation of protein content. Absorbance was evaluated at 595 nm.

## 2.6. Statistical Analysis

All data were subjected to one-way analyses of variance (ANOVA). The data analyses were performed with SPSS18.0 statistical software package for Windows. LSD multiple comparison tests were used to separate significant differences among all of the treatments at 0.05 level. SE was showed in figures and tables. C/N value was calculated from soluble total sugar and soluble protein.

# 3. Results

## 3.1. Endogenous Hormones Content

Buds and leaves showed different behaviors with respect to endogenous hormone changes at different developmental stages of *L. ruthenicum*. IAA levels in leaves and ABA, $GA_3$, and ZR levels in buds significantly increased from stage I to II ($p < 0.05$; Figure 2B,C and Figure 3A,C). From stage II to III, IAA levels of buds and ZR level of leaves increased 19.38- and 0.52-fold, respectively (Figures 2A and 3D). From stage III to IV, $GA_3$ levels of buds and leaves increased 44.50% and 17.15%, respectively (Figure 3A,B). ABA, ZR levels in buds and leaves and $GA_3$ levels in leaves significantly increased from stage IV to V ($p < 0.05$; Figure 2C,D and Figure 3B–D). From stage V to VI, the IAA, ABA, and $GA_3$

levels of buds and leaves significantly increased, while the ZR levels of buds and leaves significantly decreased at this stage (Figures 2 and 3).

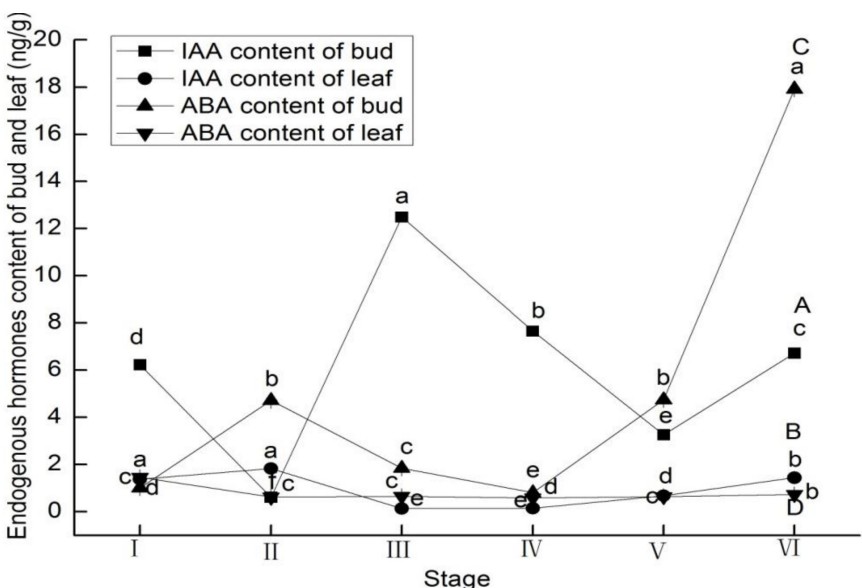

**Figure 2.** Endogenous hormones contents of indole acetic acid (IAA) in buds (A) and in leaves (B), and abscisic acid (ABA) in buds (C) and in leaves (D) of *Lycium ruthenicum* Murr. at different flower development stages. Vertical bars represent standard error (SE) of means (*n* = 3). Mean values significantly different at $p \leq 0.05$ are indicated by different letters.

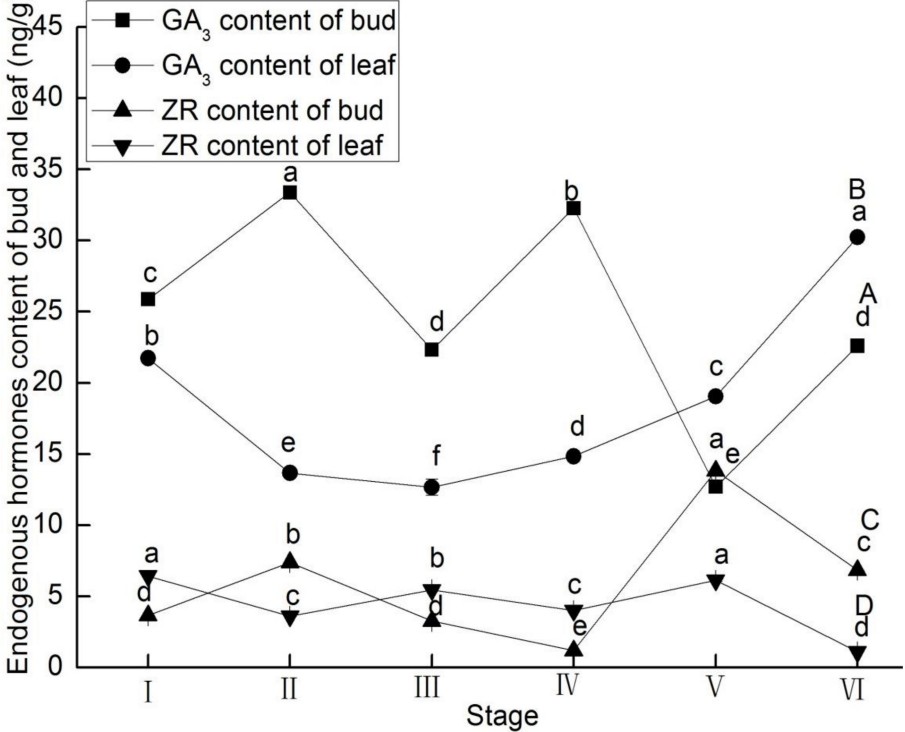

**Figure 3.** Endogenous hormones contents of gibberellic acid in buds (A) and in leaves (B), and zeatin riboside in buds (C) and in leaves (D) of *Lycium ruthenicum* Murr. at different flower development stages. Vertical bars represent standard error (SE) of means (*n* = 3). Mean values significantly different at $p \leq 0.05$ are indicated by different letters.

*3.2. Endogenous Hormones Specific Value*

In the buds, the values of ABA/IAA, ABA/GA$_3$, and ZR/IAA, ZR/GA$_3$ had a similar change tendency from stage I to IV, and significantly increased from stage I to II then decreased from stage II to IV (Figure 4). From stage V to VI, the values of ABA/IAA and ABA/GA$_3$ significantly increased, while the values of ZR/IAA and ZR/GA$_3$ significantly decreased at this stage.

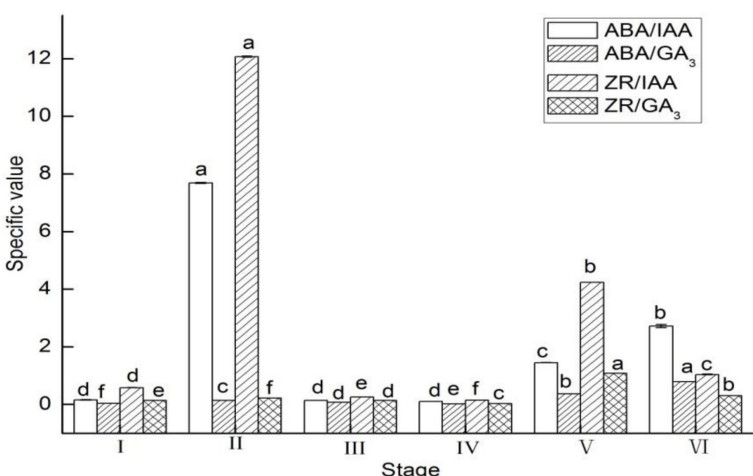

**Figure 4.** Specific value changes of ABA/IAA and ABA/GA$_3$ and ZR/IAA and ZR/GA$_3$ in buds of *Lycium ruthenicum* Murr. at different flower development stages. Vertical bars represent standard error (SE) of means. Mean values significantly different at $p \leq 0.05$ are indicated by different letters.

In the leaves, the values of ABA/IAA, ABA/GA$_3$, ZR/IAA, and ZR/GA$_3$ significantly decreased from stage I to II, and then increased from stage II to III (Figure 5). However, the values of ABA/IAA, ABA/GA$_3$, ZR/IAA, and ZR/GA$_3$ significantly decreased from stage III to VI.

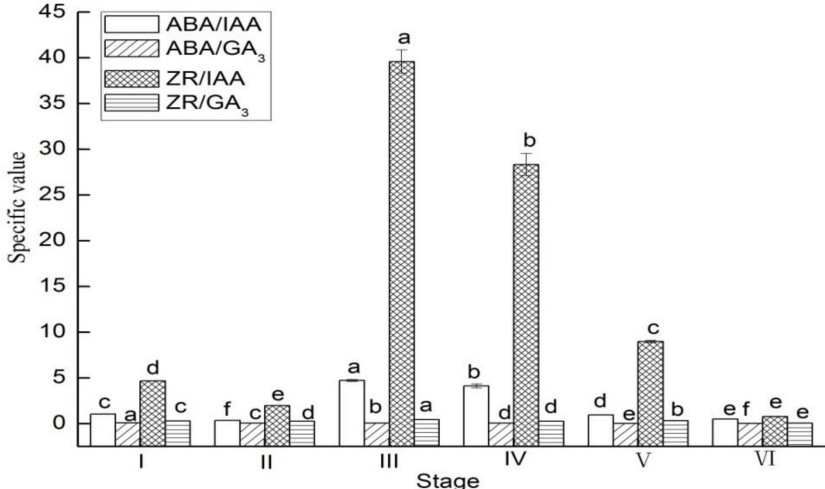

**Figure 5.** The specific value of ABA/IAA and ABA/GA$_3$ and ZR/IAA and ZR/GA$_3$ in leaves of *Lycium ruthenicum* Murr. at different flower development stages. Vertical bars represent standard error (SE) of means. Mean values significantly different at $p \leq 0.05$ are indicated by different letters.

*3.3. Carbohydrates*

Soluble sugar in the buds significantly increased from stage I to III by 8.51%, and increased 4.15-fold from stage IV to VI ($p < 0.05$; Table 1). However, soluble sugar in the leaves significantly decreased 85.89% from stage I to stage V, and significantly increased

9.27-fold from stage V to VI. Sucrose in the buds significantly increased and then decreased from stage I to VI ($p < 0.05$). The maximum sucrose value in the leaves appeared at stage III and was lowest at stage II. Starch content of bud and leaves were significantly variable during every stage ($p < 0.05$).

**Table 1.** Carbohydrates content in buds and leaves of *Lycium ruthenicum* Murr. at different flower development stages.

| Stage | Soluble Sugar (mg/mg) | | Sucrose (mg/mg) | | Starch (mg/mg) | |
|---|---|---|---|---|---|---|
| | Bud | Leaf | Bud | Leaf | Bud | Leaf |
| I | 0.047 ± 0.00 d | 0.078 ± 0.00 b | 0.021 ± 0.00 e | 0.078 ± 0.00 e | 0.012 ± 0.00 e | 0.017 ± 0.00 d |
| II | 0.044 ± 0.00 d | 0.064 ± 0.00 c | 0.024 ± 0.00 d | 0.064 ± 0.00 f | 0.009 ± 0.00 f | 0.011 ± 0.00 e |
| III | 0.051 ± 0.00 c | 0.026 ± 0.00 d | 0.070 ± 0.00 a | 0.134 ± 0.00 a | 0.207 ± 0.00 a | 0.075 ± 0.00 b |
| IV | 0.020 ± 0.00 e | 0.026 ± 0.00 d | 0.047 ± 0.00 b | 0.121 ± 0.00 b | 0.095 ± 0.00 c | 0.045 ± 0.00 c |
| V | 0.090 ± 0.00 b | 0.011 ± 0.00 e | 0.047 ± 0.00 b | 0.083 ± 0.00 d | 0.106 ± 0.00 b | 0.101 ± 0.00 a |
| VI | 0.103 ± 0.00 a | 0.113 ± 0.00 a | 0.031 ± 0.00 c | 0.113 ± 0.00 c | 0.024 ± 0.00 d | 0.015 ± 0.00 d |

Mean values significantly different at $p \leq 0.05$ are indicated by different lower-case, normal letters within a column.

### 3.4. Soluble Protein

Soluble protein contents in the buds and leaves had a similar change tendency. Soluble protein contents in the buds increased 39.28-fold from stage I to III, and decreased by 98.38% from stage III to VI (Figure 6). Soluble protein contents in the leaves increased 48.66-fold from stage Ito III, and decreased by 96.56-fold from stage III to VI.

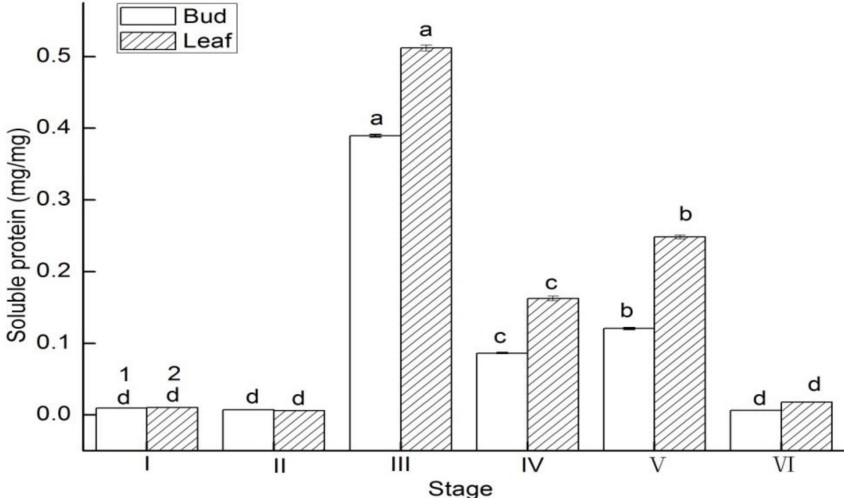

**Figure 6.** Soluble protein changes of in buds and leaves of *Lycium ruthenicum* Murr. at different flower development stages. Vertical bars represent standard error (SE) of means. Mean values significantly different at $p \leq 0.05$ are indicated by different letters.

### 3.5. C/N Value

C/N values in buds and leaves were significantly variable during flower development ($p < 0.05$).

C/N values decreased by 26% and increased 20.75% from stage I to II and between stages V to VI, respectively, in buds (Figure 7). C/N values increased by 39% and 143.34% from stage I to II and between stages V to VI, respectively, in leaves.

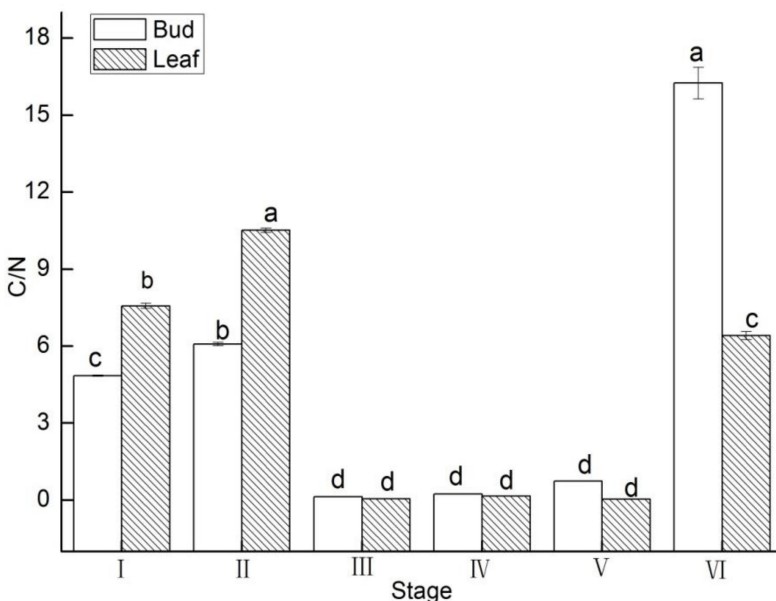

**Figure 7.** C/N value changes of in buds and leaves of *Lycium ruthenicum* Murr. at different flower development stages. Vertical bars represent standard error (SE) of means. Mean values significantly different at $p \leq 0.05$ are indicated by different letters.

## 4. Discussion

Flower bud differentiation is a complex process, and is affected by multiple factors. Endogenous hormone content and their balance are important effects for flower bud differentiation [15–17]. The relation of balance is mutual restraint and promotion, and control the metabolism nutrient substance to adjust and control flower bud differentiation. IAA is the main factor responsible for inflorescence stalk elongation [18]. IAA was also controversial, as both positive and negative effects of IAA have been reported [19]. Our study showed that IAA content of buds significantly decreased from flower bud pre-differentiation to late flower differentiation. The results suggest that floral induction need lower IAA levels in *L. ruthenicum*. Similar results have been reported for *Chrysanthemum* [20]. However, IAA content of leaves significantly increased at this stage. The results showed a relation of IAA transportation from morphological upper buds to morphological lower leaves at this stage. Similar results have been reported for *Chrysanthemum* [20]. At the squaring stage, the IAA content of buds significantly increased compare with flower bud differentiation stage. The results indicated that squaring need higher IAA levels in *L. ruthenicum*. Similar results have been reported in apricot [21]. ABA is essential in flowering and plays an important role in flower development [22]. The ABA content of buds significantly increased from flower bud pre-differentiation to late flower differentiation. The results suggest that ABA could promote flower differentiation of *L. ruthenicum*. ABA content of buds and leaves maintained the higher level from open stage to senescent stage in *L. ruthenicum*. It indicated that ABA can promote the flower opening and senescence in *L. ruthenicum*. Some research showed that GA$_3$ had an inhibitory effect to florescence [23–27]. However, other research found that lower concentration GA$_3$ can promote floral development in terminal buds of apple trees [8]. Our research showed that GA$_3$ content of buds significantly increased from flower bud pre-differentiation to late flower differentiation. The results suggest that the higher GA$_3$ could promote the flower differentiation of *L. ruthenicum,* similar to olive [7]. The ZR content in buds significantly increased from flower bud pre-differentiation to late flower differentiation. The results showed that ZR could promote flower bud differentiation of *L. ruthenicum,* similar to potatoes [28]. At the open stage, the ZR content reached the maximum in buds and leaves of *L. ruthenicum*. Flower formation is the result of all kinds of hormone interaction. Our results showed that for all hormone levels analyzed, their proportion changed at different development stages of *L. ruthenicum*.

ABA/IAA in the buds gradually increased before the open stage. The increase indicated that ABA/IAA in buds benefits flower bud differentiation and flower development in *L. ruthenicum.* Increased ABA/GA$_3$ values in the buds and leaves were in accordance with the flower development in *L. ruthenicum.* Research in Jujube [25] and *Lycoris radiata* [29] obtained similar results. ABA/GA$_3$, ABA/IAA, and ZR/IAA in the buds were higher than in the leaves. This finding indicated that hormone levels and proportions in the buds had an important effect on *L. ruthenicum* flower development. Nutrient substance accumulation is the basis of the flower bud differentiation. Carbohydrates have an important contribution to the flower bud differentiation. Priestly [30] suggested that a high carbohydrate level alone does not promote flowering. In our research, the higher soluble sugar and sucrose content in the buds and leaves were in accordance with the flower bud differentiation of *L. ruthenicum.* Similar results have been reported in olive [7] and *Olea europaea* var [31]. Florescence and senescent process of plant need to consume abundant respiratory substrate, and need enough energy to push florescence and senescent. Our research showed that the increasing of soluble sugar improved the energy basis to florescence and senescent. Nitrogen is the main nutrient element of plant growth. The soluble protein levels of buds and leaves had the same variation tendency in *L. ruthenicum.* The soluble protein levels were unaltered from flower bud pre-differentiation to late flower differentiation. At the squaring stage, soluble protein levels of buds and leaves reached maximum in *L. ruthenicum.* At the open and senescent stage, soluble protein levels of buds and leaves significantly decreased in *L. ruthenicum.* Research on *Nicotiana plumbaginifolia* Viv [9] obtained similar results. The decrease was associated with improved proteolytic cleavage or decreased protein biosynthesis and was regarded a prerequisite to the open stage in various flowers [32,33]. The C/N ratios reflect the basic state of carbon and nitrogen metabolism. The C/N ratios of buds and leaves were higher at the flower bud differentiation stage of *L. ruthenicum*, because carbohydrate can improve vascular sap concentration and provided important nutrient substances to flower bud differentiation stage. The C/N ratios in the buds and leaves significantly increased from the open stage to senescent stage. This result showed that the carbohydrate metabolism enhanced from the open stage to senescent stage and that the nitrogen metabolism reduced from the open stage to senescent stage of *L. ruthenicum.*

## 5. Conclusions

Our current study demonstrated that a higher ABA, GA$_3$, and ZR content and ABA/IAA, ABA/GA$_3$, ZR/IAA, and ZR/GA$_3$ values in buds could contribute to the promotion of the flower bud differentiation of *L. ruthenicum.* Higher ABA/IAA, ABA/GA$_3$, ZR/IAA, and ZR/GA$_3$ values in leaves were correlated to flower development in *L. ruthenicum.* Higher IAA, ABA, and GA$_3$ values in buds and leaves correlated with the senescent of flowers in *L. ruthenicum.* Higher soluble sugar and sucrose content in the buds and leaves was beneficial to the flower bud differentiation of *L. ruthenicum.* The increase of soluble sugar improved the energy basis to florescence and senescent.

**Author Contributions:** Y.G. designed experiments, performed data analysis, and wrote the paper; L.A. performed data analysis and field detection; H.Y. and M.Y. performed data analysis and revised the paper. All authors have read and agreed to the published version of the manuscript.

**Funding:** This work was supported by programs of the National Natural Science Foundation (Grant No. 32060334; Grant No.31660193).

**Conflicts of Interest:** The authors declare no conflict of interest.

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
