# Peer review of "Endogenous Hormones and Biochemical Changes during Flower Development and Florescence in the Buds and Leaves of Lycium ruthenicum Murr"

_forests, doi:10.3390/f13050763_

Round 1

Reviewer 1 Report

The manuscript is in need to have thorough revision and editing. 

There are many mistakes which need to be corrected before considering for another review.

Author Response

QUESTION: page 1, line25, 26

ANSWER: We have modified the sentence. Please see new manuscript.

QUESTION: page 2, line34

ANSWER: We have added “and”. Please see new manuscript.

QUESTION: page 2, line52

ANSWER: We have modified the references. Please see new manuscript.

QUESTION: page 2, line54

ANSWER: We have deleted “and”. Please see new manuscript.

QUESTION: page 2, line64

ANSWER: We have d modified the sentence. Please see new manuscript.

QUESTION: page 3, line81,82,83,84

ANSWER: We have modified the sentence. Please see new manuscript.

QUESTION: page 3, line86,87,88

ANSWER: We have d modified the sentence. Please see new manuscript.

QUESTION: page 3, line90,91,92,93

ANSWER: We have d modified the sentence. Please see new manuscript.

QUESTION: page 3, line95

ANSWER: Around have modified “the”. Please see new manuscript.

QUESTION: page 3, line99

ANSWER: We have modified the references. Please see new manuscript.

QUESTION: page 3, line101,102

ANSWER: We have modified the sentence. Please see new manuscript.

QUESTION: page 4, line112,116,119,124,127,128,136

ANSWER: We have modified the sentence. Please see new manuscript.

QUESTION: page 6, line152,153,165

ANSWER:We have modified the sentence. Please see new manuscript.

QUESTION: page 7, line175, 193,194,195

ANSWER:We have modified the sentence. Please see new manuscript.

QUESTION: page 8, line210,211,213,214,221,222,223,224

ANSWER:We have modified the sentence. Please see new manuscript.

QUESTION: page 9, line229,230,237,239,240,246,263

ANSWER:We have modified the sentence. Please see new manuscript.

QUESTION: page 10, line271,272

ANSWER:We have modified the sentence. Please see new manuscript.

Reviewer 2 Report

In this work the content of different hormones and metabolites was measured during the flower development of Lycium ruthenicum. After the establishment of six stages in the flower development, the authors performed the content measurement of four hormones (IAA, ABA, GA3 and ZR), carbohydrates and soluble proteins, and established the ratio in some of those. The results showed changes in those hormones and metabolites during the stages of flower development.

The commets to the manuscript:

I suggest to improve the English writing in all the manuscript to permit an easy reading and understanding. 

The results are interesting but the description difficults their comprehension; it is necessary to indicate the number of samples analyzed per stage, the figure legends and the labels in the graphs must be corrected, etc.

My major concern is about some statements that are presented in the Discussion and Coclusions sections that are not according, in my opinion, to the presented results.

My suggestions and comments are mentioned in the article file.

Author Response

QUESTION: page 1, line6,7,15,20,22,23

ANSWER:We have modified the sentence. Please see new manuscript.

QUESTION: page 1, line24

ANSWER: We have modified the soluble. Please see new manuscript.

QUESTION: page 1, line25,26,27

ANSWER:We have modified the sentence. Please see new manuscript.

QUESTION: page 2, line33,42,44,45,49,52,64,57,63,67,68,70,71

ANSWER:We have modified the sentence. Please see new manuscript.

QUESTION: page 3, line 82,84,86,87,88,91,92,93

ANSWER:We have modified the sentence. Please see new manuscript.

QUESTION: page 4, line114, 115, 116, 119, 120, 121, 122, 123, 131, 132, 133, 134,135,136,137,138

ANSWER:We have modified the sentence. Please see new manuscript.

QUESTION: page 5, line142,143,144,146,148

ANSWER:We have modified the figure and the sentence. Please see new manuscript.

QUESTION: page 6, line152,153,154,156

ANSWER:We have modified the sentence and figure. Please see new manuscript.

QUESTION: page7, line174,175,178,189,193

ANSWER:We have modified the sentence. Please see new manuscript.

QUESTION: page8, line196, 197, 203, 204, 206, 210, 211, 212, 213, 214, 217, 218, 223

ANSWER:We have modified the sentence. Please see new manuscript.

QUESTION: page9, line225, 226, 228, 229, 230, 234, 235, 236, 240, 243, 245, 249, 250,253,263,

ANSWER:We have modified the sentence. Please see new manuscript.

QUESTION: page10, line268,269,271,272,275,277,278,280

ANSWER:We have modified the sentence. Please see new manuscript.

Round 2

Reviewer 1 Report

Accept 

Reviewer 2 Report

I appreciate so much the authors took in account all my suggestions and corrections to the text. And in this regard my major concern is about the English grammar. The correcctions were only by the suggestions I did, but my native language is not  English. In order to improve the manuscript I advice to be revised by an English editor.

I did some new minor corrections, but one of them is about the results described from graph in figure 7. Please, the concern about this is discussed in the enclosed revised document.
